# The Regulation and Immune Signature of Retrotransposons in Cancer

**DOI:** 10.3390/cancers15174340

**Published:** 2023-08-30

**Authors:** Maisa I. Alkailani, Derrick Gibbings

**Affiliations:** 1College of Health and Life Sciences, Hamad Bin Khalifa University, Qatar Foundation, Doha P.O. Box 34110, Qatar; 2Department of Cellular and Molecular Medicine, Faculty of Medicine, University of Ottawa, Ottawa, ON K1H 8M5, Canada; gibbings@uottawa.ca

**Keywords:** transposable elements, mobile genome, insertions, tumorigenesis, immunity, type I IFN, jumping genes

## Abstract

**Simple Summary:**

A tiny human sample is enough to uncover the complete genome sequence of that individual with the advances in biomedical technologies and data analysis. Jumping genes constituting about half of the human genome, have been implicated in cancer and predisposition to inflammatory reactions. Inflammation may restrict the activity of these genes and reduce the tumor burden. This article summarizes related literature on factors regulating jumping genes and discusses their immune-related evidence made available by genome-wide studies.

**Abstract:**

Advances in sequencing technologies and the bioinformatic analysis of big data facilitate the study of jumping genes’ activity in the human genome in cancer from a broad perspective. Retrotransposons, which move from one genomic site to another by a copy-and-paste mechanism, are regulated by various molecular pathways that may be disrupted during tumorigenesis. Active retrotransposons can stimulate type I IFN responses. Although accumulated evidence suggests that retrotransposons can induce inflammation, the research investigating the exact mechanism of triggering these responses is ongoing. Understanding these mechanisms could improve the therapeutic management of cancer through the use of retrotransposon-induced inflammation as a tool to instigate immune responses to tumors.

## 1. Background

“You just know sooner or later, it will come out in the wash, but you may have to wait sometime.” Dr. Barbara McClintock conveyed this statement upon receiving the Nobel Prize recognizing her discovery of transposable elements (TEs) [1]. TEs are mobile DNA sequences that can move from one genomic location to another in a process called “transposition” [2]. Transposition in the genome is facilitated by one or more proteins encoded by a TE [3]. In this review, we shed light on the regulatory mechanisms affecting the active classes of TEs and their immunological impact on human cancer using evidence from recent genome-wide studies. As illustrated in Figure 1, TEs are categorized into two broad classes: DNA transposons and retrotransposons, based on their transposition intermediate and mobility mechanisms [3]. DNA transposons are sequences that use element-encoded transposases to move from one genomic location to another by a cut-and-paste mechanism [3]. A retrotransposon element inserts into a new genomic location by a copy-and-paste mechanism using an RNA intermediate [3,4]. This article’s focus is on retrotransposons since there is no evidence of DNA transposon insertion into the human genome in the last 37 million years [5].

According to the presence or absence of long terminal repeats (LTR) in their sequences, retrotransposons are subdivided into LTR- and non-LTR-containing elements [3]. Human endogenous retroviruses (HERVs) are autonomous protein-encoding LTR-containing elements [6]. Most HERV elements are non-functional due to accumulated mutations or internal recombination, resulting in solitary LTRs [7]. However, the evidence suggests the recent insertion of HERV elements within the human population in polymorphic loci [8]. Retrotransposons lacking LTR include long interspersed elements (LINEs) and short interspersed elements (SINEs) [6]. Of these, the most active elements retrotransposing in the human genome include autonomous LINE-1 (L1) from the LINEs and non-autonomous *Alu* from the SINEs (reviewed in [3]). L1 has two promoters (sense and antisense) to transcribe three different open-reading frame (ORF) regions. The sense promotor transcribes ORF1 and ORF2 [9,10]. At the same time, the antisense promoter transcribes a primate-specific ORF (ORF0) in the opposite orientation to that of L1 [11]. ORF1 encodes a 40 kDa protein (ORF1p) with a nucleic acid chaperone and RNA binding activities [12]. ORF2 encodes a 150 kDa protein (ORF2p) that has endonuclease (EN) and reverse transcriptase (RT) activities [13,14]. The *Alu* elements are primate-specific retrotransposons, with the most recent amplification in lineages attributed to a series of *Y* subfamilies (*Ya5* and *Yb8* dominate in humans) [15]. Each *Alu* element comprises two dimers ancestrally derived from the 7SL RNA and separated by a short polyA sequence. A longer polyA tail occupies its 3′ end [15]. *Alu* elements do not encode proteins; instead, they hijack L1 proteins to mediate their retrotransposition [15], which occurs through the life cycle of L1, starting with the transcription of L1 mRNA from its genomic copy [3]. L1 RNA is exported to the cytoplasm, where the ORF1p and ORF2p proteins are translated [16]. These proteins bind the retrotransposon RNA (L1/*Alu*) to form a ribonucleoprotein particle (RNP) [17,18]. The RNP is imported into the nucleus to facilitate L1/*Alu* retrotransposition via two distinct pathways [19]. The canonical pathway is called target-primed reverse transcription (TPRT) [20,21], in which the L1 EN activity produces a nick at a target site in the genomic DNA [13,22]. It preferentially cuts DNA at the consensus sequence 5′-TTTT/A-3′ or its variants [22]. Then, using the retrotransposon RNA as a template, the L1 RT moiety extends the unbound 3′-OH group from DNA to begin reverse transcription, starting within the polyA tail of the retrotransposon RNA [3,22]. Retrotransposition can also occur via an endonuclease-independent pathway or non-classical L1 insertion. Endonuclease cleavage is not required in this pathway, and the reverse transcription is initiated at pre-existing DNA break regions [23,24].

De novo retrotransposon insertions can occur in exons, introns, or the regulatory regions of the genome, disrupting their function, providing new promoter and enhancer regions, and contributing to disease [25,26]. These insertions can exert deleterious, “disruptive,” or beneficial “exaptation” effects on the host [27]. Retrotransposition in introns can affect the splicing process by different mechanisms [28,29]. It can provide alternative (donor or acceptor) splice sites, cause exonization (a process by which genes acquire new exons from intronic DNA sequences), or promote exon skipping [28,29]. Alternative splicing and exon-acquisition events of the CHRM3 gene, a muscarinic acetylcholine receptor family member, are examples of TE integrations into the host genome that are naturally selected and conserved over generations [30]. About 62% of exonizations in the human genome are *Alu*-derived [31]. The insertion of *Alu* into one of the *Factor VIII* gene introns resulted in exon skipping and the consequent onset of hemophilia A [32]. Table 1 outlines the mechanisms by which retrotransposons can impact genomic structure and function.

## 2. Regulation of Retrotransposons and Their Association with Tumorigenesis

Reports have demonstrated that retrotransposon expression and activity occur primarily in cells associated with the germline, with little expression in most somatic tissues under physiological conditions [51]. L1 retrotranspositions can occur during early human embryonic development [52]. They were identified in neuronal precursor cells [53] and have been observed in various cell lines when a tagged L1 construct was employed. However, limited data are available on whether retrotranspositions occur in normal somatic adult tissues other than the brain [54,55,56]. A few findings indicated that somatic insertions in hepatocytes and the esophagus, stomach, and colon may have occurred during embryogenesis [57,58,59]. This lack of evidence could be related to the somatic insertions occurring in a few cells within the tissue that are challenging to identify in whole-tissue sequencing.

De novo somatic insertions were identified in different tumor tissues of epithelial origin at varying frequencies [60,61]. These insertions are characterized by the fact that they have more 5′ truncations and exist with less dependence on L1-encoded EN cleavage than germline insertions [62]. Retrotransposon activity was associated with tumorigenesis in the early observations of Miki et al., who detected that a novel L1 insertion impacted the tumor suppressor *APC* in colon cancer but not in normal colon tissues from affected individuals [63]. More than two decades later, another L1 insertion was found to disrupt the other allele of the *APC* gene, which contributed to colon tumorigenesis [64]. *ST18* (suppression of tumorigenicity 18) and *PTEN* genes were other tumor suppressors interrupted by new L1 insertions in hepatocellular carcinoma and endometrial cancer, respectively [65,66]. There is much evidence to show that retrotransposons are crucial contributors to tumorigenesis, especially with the global epigenetic dysregulation that characterizes tumorigenesis [67].

By introducing the high-throughput L1-sequencing assay, Iskow et al. could identify a hypomethylation signature that characterized lung tumors, which made them more L1-permissive and had a higher frequency of L1 somatic insertion than the brain tumors included in that study [68]. Tumors of epithelial origin, such as colorectal, prostate, and ovarian cancers, showed more pronounced L1 activity than the brain and blood cancer types with the performance of single-nucleotide resolution analysis of TE insertions in whole-genome sequencing datasets [61]. In agreement with these findings, colorectal and lung cancers were the most frequently affected by L1 somatic insertions exhibiting hypomethylated promotors by tracking down the L1 insertion sources via the identification of 3′ transductions [69]. The preference for retrotransposon activity in specific tumor types could be related to a range of transcription factors activated in specific cell types over others. The activation of transcription factors in epithelial tumors might modulate retrotransposon expression and activity. For example, epithelial tumors such as breast, colorectal, prostate, and cervical cancers are characterized by Oct1 (Octamer transcription factor 1, POU2F1) protein upregulation [70,71,72,73]. Oct1 controls stem cell phenotypes in normal and tumor cells [73]. In epithelial cells, high Oct1 protein expression was spatially correlated with stem cell niches and the increased expression of stem cell markers such as ALDH1 [73]. Transcription factors like Oct1 may play a role in epithelial cell de-differentiation into a more stem-like phenotype [74]. These cells may be more disposed to L1 retrotransposition than other populations of cancer cells [74].

Several transcription factors were demonstrated to regulate the transcription of retrotransposons by binding their promoters. These factors include YY1, RUNX3, p53, Oct4, Sox2, Nanog, KLF4, MYC, and CTCF [75,76,77,78,79,80,81,82,83]. Although the L1 5′UTR promoter region is prone to higher mutation rates than the L1 ORF regions, the evolutionary analysis showed conservation in the transcription factor binding sites among human-specific L1 elements [84]. The transcription factors regulating retrotransposon expression are not isolated from other regulators that modulate retrotransposon activity in the cell. Each of these regulators is a part of different pathways that make up an interconnected network of factors controlling retrotransposon expression and activity.

Retrotransposons have long been considered genomic threats to somatic cellular functions and are under control mechanisms that restrict their activity [85]. These regulation mechanisms sometimes fail in cases of age or disease [85]. The factors restricting retrotransposons fall into one of two categories: cytoplasmic or nuclear—most factors acting in the cytoplasm limit the retrotransposon’s expression by post-transcriptional mechanisms. The suppressing nuclear factors either restrict the transcription of retrotransposons or interfere with their genomic integration (see Table 2). These factors (being cytoplasmic or nuclear) are illustrated in Figure 2 based on the retrotransposon’s life cycle.

## 3. Retrotransposons in Cancer from a Genome-Wide Perspective

Recent advances in bioinformatics tools have paved the way for studying retrotransposons. It is a significant challenge to precisely determine their insertion sites using standard DNA sequencing technologies. This difficulty can be related to the retrotransposon sequence characteristics or the available data quality. The L1 sequence, for example, differs among genomic copies in terms of the polyadenylation signal and 3′ UTR, with most copies being 5′ truncated [104,105,106]. Most available whole-genome sequencing (WGS) data consist of single- or paired-end short reads of about 100–250 nt in length [107]. Using these reads to detect 6000 kbp L1 insertions requires methods to identify the sequences overlapping TE elements and new genomic locations. Filters and measures are needed to reduce the number of false-positive insertions detected while maintaining reasonable sensitivity in detecting new TE insertion events [107].

Large-scale sequencing projects include data from thousands of individuals deposited in public databases such as The Cancer Genome Atlas (TCGA) and the International Cancer Genome Consortium (ICGC). In addition, bioinformatics tools and pipelines have facilitated the comprehensive detection and analysis of retrotransposons in cancer [108]. The available tools that accelerate research in the TE field can range from data repositories to insertion detecting tools and strategies to investigate the TEs’ biological impacts. Databases such as RepBase Update and the European database of L1HS retrotransposon insertions (EUL1Db) were developed as repositories focused on assembling TE consensus sequences with the reference genome and identifying common polymorphic TE insertions [109,110].

Two factors are required to identify TE polymorphisms in an individual sequenced genome: an available reference genome and the annotated TE sequences in that genome; both are made accessible in public databases. TE polymorphisms are detected using reads that span the borders of retrotransposons and new genomic locations in the search for retrotransposons not yet included in the reference sequence [107]. Some identified polymorphic insertions were linked to diseases such as hemophilia [37] and Rett syndrome [111]. Many TE detection software tools have been developed to identify germline and somatic TE insertions using short-read sequencing, as in the TCGA [108]. Short reads do not frequently span the entire interval affected by retrotransposon-mediated genomic rearrangement [107]. Therefore, computational tools were developed to utilize up to three strategies in detecting TE insertions: inference from discordant read pair (DRP) mapping; clustering of split reads (SR); and sequence re-alignment through the identification of TE-specific motifs [112]. DRP methods detect a pair of reads from the same TE insert whose alignment to the reference sequence has an orientation or distance that differs from the expected range [113]. No identification of exact junctions between TEs and the reference genome is possible using DRP methods alone [107].

On the other hand, the SR methods detect reads that map partially with the surrounding genome and partially in a TE sequence [113] (Figure 3A). Non-reference SRs are clipped to align with the reference sequence and can be used to identify the junctions between the TE and reference genome sequence [107]. Therefore, SR strategies provide a higher positional accuracy by identifying the junction between the TE and host sequence. DRP strategies, on the other hand, offer higher sensitivity, providing more reads to support TE insertions [108]. However, another strategy is required to refine the DRP mapping by requiring an SR- or TE-specific motif detection to exclude TE-unrelated rearrangements [107]. In the TE-specific motif detection strategy, tools were developed to identify insertions by looking for common TE signatures, such as target site duplications (TSDs) flanking most TE insertions, long stretches of poly (A) tails, and 3′ transduction in L1-mediated insertions [108].

The TE field advances have been extended to offer tools that predict the impacts of TEs on gene regulation, such as measuring the overlap with other genomic regions, looking for associations with transcription regulation datasets, or considering signs for negative or positive selection [108]. In searching for active TEs and studying the effect of these elements on the expression of nearby genes, alignment tools were developed, such as RepEnrich [114] and SQuIRE [115]. These tools are designed to identify the differential expression analysis of TEs in chromatin immunoprecipitation (ChIP) sequencing and/or RNA sequencing data [114]. The RepEnrich tool creates a series of contiguous segments representing all TE instances of each TE subfamily annotated in the TE repository (e.g., Repbase, Figure 3B) [114]. These series are then used to identify reads that map only to one subfamily of TEs, such as L1HS (Figure 3C). The reads identified using this tool can be described as unique to a particular subfamily in the genome. The SQuIRE tool quantifies the TE subfamily expression and performs differential analyses on TEs and genes at the locus level [115] (Figure 3D). As summarized in Table 3, genome-wide research follows one of two strategies used to study retrotransposon activity in cancer: targeted resequencing assays and bioinformatics analysis of WGS or whole-exome sequencing (WES) data.

## 4. Immune Signature of Retrotransposons in Cancer

Most of the (above-mentioned) genome-wide studies were focused on identifying new insertions and characterizing their effect on tumor-modulating genes. There is also a growing interest in identifying the factors controlling retrotransposon RNA expression or the factors triggered by its activation, such as the emerging data demonstrating that retrotransposon activation can be immunogenic and may instigate IFN and apoptosis signaling [118,121,122,123,124].

Tumors with high immune activity, such as those associated with the Epstein–Barr virus (EBV) infection, demonstrated a low number of L1 insertions [118]. Reports also indicate high retrotransposon activity in head and neck squamous cell carcinoma (HNSCC) patients. The overexpression of retrotransposons in HNSCC was shown to be associated with robust DNA CpG demethylation of tumor tissue [125]. A high expression of the long terminal repeat (LTR) retrotransposon HERVs in HNSCC cases was accompanied by high cytolytic effectors, which correlated positively with cytolytic immune activity [126]. This activity could be related to the oncogenic human papillomavirus (HPV), whose infection is among the etiological factors contributing to a subset of HNSCC tumors. HPV-positive cases often present with better outcomes [127] and are less likely to have *TP53* mutation [66]. These tumors have also demonstrated less retrotransposon somatic insertions (i.e., activity) [66]. The examples above suggest the involvement of a defense mechanism against retrotransposons resembling antiviral actions.

Many retrotransposon regulation mechanisms are similarly used to protect cells from exogenous viral infections. When nucleic acids of foreign origin are detected by endosomal or pattern recognition receptors (PRR), an IFN-driven immune response is initiated to eliminate the affected cell populations [128]. The cell is equipped with a heterogeneous group of PRRs that includes but is not limited to Toll-like receptors (TLR3, TLR7, TLR8, and TLR9); the RNA sensors RIG-I (retinoic acid-inducible gene I), MDA5 (melanoma differentiation-associated protein 5), and LGP2 (RIG-I-like receptor LGP2); and the DNA sensors cGAS (cyclic GMP-AMP synthase) and AIM2 (absent in melanoma 2) [129].

Specific criteria, including location, nucleic acid sequence pattern, and threshold quantity, determine which nucleic acid each PRR senses [129]. TLR nucleic acid binding domains face the lumen of endosomal compartments, and the other PRRs are present in the cytoplasm [129]. TLR3 binds dsRNA of >40 bp size; TLR7/8 bind fragmented RNA with unmodified nucleosides; and TLR9 binds ssDNA of >11 nt size with a high affinity to the unmethylated cytosine CpG motif [129]. RIG-I binds >20 bp dsRNA with blunt end conformation; MDA5 binds >1–2 Kb dsRNA; cGAS binds dsDNA of >20–40 bp size; and AIM2 binds dsDNA of >50–80 bp size [129]. The quantity of detected nucleic acid can be affected by the increased supply that causes the accumulation of nucleic acids and the defective mechanisms of their clearance.

The failure of one or more of the (above-described) retrotransposon regulatory mechanisms (due to aging, tumorigenesis, or autoimmune disease) can result in retrotransposon activation. This activity promotes dsRNA or dsDNA (sequences of different sizes and motifs) release into the cytoplasm and their detection by cGAS or MDA5, respectively [123,124,130]. Most ADAR-mediated A-to-I RNA editing sites are found in close proximity to retrotransposons. Upon the depletion of ADAR1 in conditions such as Aicardi–Goutières syndrome and some cancers, unedited endogenous RNAs trigger a chronic type I IFN response via MDA5 facilitated by the LGP2 RNA sensor [131]. The activation of L1 during cellular senescence triggered the release of L1 dsDNA in the cytoplasm and promoted type I IFN responses and sterile inflammation [122].

In addition to the evidence summarized in Table 4 below, many examples suggest the retrotransposon activation of innate immune response in cancer. By analyzing TCGA RNA sequencing data, specific HERV elements were highly enriched in tumor samples compared to their normal counterparts, and this enrichment was associated with an increased immune response [126]. Another piece of evidence showed that cytosolic ssDNA and dsDNA in several tumor cell lines were mainly retrotransposon-derived and associated with the cGAS-activated STING and type I IFN response [130]. Activating HERV expression using DNMT inhibitors (DNMTi) in cancer cells triggered cytosolic dsRNA release, and MDA5 stimulated immune response [124]. In addition, expressing ERV sequences in TLR3, TLR7, and TLR9 triple-deficient mice failed to induce a sufficient immune response, resulting in their development of T-cell acute lymphoblastic leukemia and their early death [132]. Blood samples from individuals with the autoimmune disease SLE, systemic lupus erythematosus, were enriched in *Alu* RNA associated with high levels of type I IFN response [133]. Although the triggers of retrotransposon activation in the disorders mentioned above may differ, their induction of TEs is likely to be the cause of the IFN responses as a means of protection. A feedback loop may be generated to inhibit L1 activity, as suggested by specific interferon-stimulated proteins directly interacting with its encoded ORF1p [134].

Tumor-specific characteristics may alter the tumor microenvironment and play a role in retrotransposon expression and its associated immune response. TP53, for example, has immunomodulatory roles, and its dysfunction associates with immunosuppression [126,146], which is consistent with the evidence of gastrointestinal tumors with *TP53* mutations showing low immune activity and higher loads of L1 insertions than tumors with wild-type *TP53* [118]. Also, evidence from colon cancer shows that in response to viral infection in cells, TP53 induces an IFN-dependent antiviral response by activating IFN-stimulated genes [147]. Another piece of evidence showed that *TP53* cooperates with DNA methylation to maintain the silencing of SINEs and other non-coding RNAs [148]. The *TP53*-deficient cells in this study exhibited high SINE element expression accompanied by a high type I IFN response [148]. However, not all tumors exhibit the same type of *TP53* mutation, and not all mutations result in TP53 protein deficiency [149]. *TP53* mutation can contribute to tumorigenesis by losing TP53 function and gaining mutant functions [149]. Whereas frequent *TP53* loss of function mutations in basal-like breast cancer could increase retrotransposon expression and the associated IFN response, the *TP53* gain of function mutations in high-grade serous ovarian tumors could reduce retrotransposon expression and its associated IFN response [86].

Apart from *TP53*, gastrointestinal tumors had strong associations between retrotransposons and TLRs or IFN-induced mRNAs, which was not the case in breast and ovarian cancers [86,118]. Also, IFNε, which is hormonally regulated and expressed in the cells of reproductive organs [150], presented high associations with retrotransposon expression in breast and ovarian cancers [86]. It could be that because retrotransposons contain several binding motifs for estrogen response elements (ERE) [151], they may play a role in IFNε expression in the tumors of reproductive organs. Therefore, the effect of IFN on retrotransposons could be related to the hormone-regulated microenvironment and might be tumor type-specific. The context-dependent IFN signaling associated with the ER+ and ER-negative breast cancer subtypes, which impacts their response to therapy and overall outcomes, reinforces the above notion [152]. It could be interesting to extend these experiments to identify the levels of retrotransposon expression among ER+ and ER- breast tumors. The examples above support the assertion that the tumor type and specific characteristics could affect the retrotransposon’s expression and linked immune response. These variabilities should be considered when studying the retrotransposon’s activity in different types of cancer.

## 5. Therapeutic Opportunities for Retrotransposon Activity in Cancer

Throughout their evolutionary timeline, significant retrotransposon-related activities at the genomic and cellular levels have been attributed to their RT [14]. However, retrotransposon genomic insertions in cancer have drawn considerable attention beyond the attention given to retrotransposon RT activity [153]. RT activity was shown to increase during tumorigenesis. Anti-retroviral non-nucleoside reverse transcriptase inhibitors (NNRTIs), such as efavirenz and nevirapine, reduced RT activity significantly by inducing conformational changes in the enzyme [154,155]. The NNRTIs reduced tumor growth by decreasing cellular proliferation and promoting differentiation [156,157]. The effect of inhibiting RT using NNRTIs was similar to that of the L1 siRNA suppressing effect; therefore, they were assumed to target L1 activity [158]. Other lines of evidence suggest that another class of RT inhibitors, nucleoside reverse transcriptase inhibitors (NRTIs), are capable of inhibiting L1 activity and having anticancer effects in cells [159,160]. This evidence suggests that L1-encoded RT is a potential marker for diagnostic purposes and a potential target for therapeutic intervention. However, further work is still required to understand the exact mechanism of the observed effect of RT inhibitors on cancer [161]. Although both NRTIs and NNRTIs could inhibit cancer cell growth, only NRTIs inhibited telomerase RT in vitro [162], which may suggest a mechanism related to L1 RT particularly to affect cancer growth. 

Among the mechanisms that activate retrotransposons, demethylating agents such as DNMTi act by releasing the epigenetic restriction placed on retrotransposons [123,125,163]. Activating various TE classes in glioblastoma cells triggered type I and II IFN responses [125]. TE-derived peptides were processed and presented on MHC class I molecules that activated adaptive immunity [125]. Activation of HERVs resulted in a viral mimicry response of dsRNAs, inducing the MDA5/MAVS RNA recognition pathway and the downstream activation of interferon response factor 7 (IRF7) [123]. Recent evidence (based on TCGA data analysis and in vitro DNMTi treatment of ovarian cancer cells) suggested that high HERV expression in patients was associated with better survival and correlated with the infiltration of cytotoxic T cells [164]. The use of DNA-hypomethylating agent 5-azacitidine (AZA) in colon and ovarian cancer cell models was associated with the increased expression of HERV and L1 RNA [124,165]. HERV expression was linked to regulatory T cell tumor infiltrates and predicted cytolytic activity in AZA-treated cells [165].

In contrast, L1 expression correlated with TP53 status and predicted AZA drug sensitivity [165]. A dinitroazetidine derivative (RRx-001), another hypomethylating drug less toxic than AZA, is currently in phase II clinical trials [166]. RRx-001 induced antitumorigenic effects by activating the expression of HERV and IFN-responsive genes [166]. Similarly, treating colon cancer cells and tumor organoids with another derivative of a hypomethylating agent (5-aza-2′-deoxycytidine) was sufficient to induce a growth-inhibiting immune response by triggering retrotransposon expression [123,163]. Interestingly, the combination of DNMTi and HDACi selectively induced LTR retrotransposons more efficiently than using each drug individually [167]. The treatment-activated TSS of LTR elements induced them de novo from non-annotated TSS [167]. This activation resulted in chimeric products with predicted immunogenic functions [167].

In addition, some targeted cancer therapeutics and chemotherapeutic agents were shown to activate retrotransposon expression in cancer cells [121,168]. Cyclin-dependent kinases 4 and 6 (CDK4/6) inhibitors repressed DNMT1 and caused activation of repeat elements, including retrotransposons in breast cancer [168]. This activation promoted cytotoxic T-cell-mediated clearance of tumor cells and increased tumor immunogenicity [168]. However, some cells within a heterogeneous cancer population may develop adaptation mechanisms to survive the challenging tumor microenvironment conditions [121]. These cells could modulate retrotransposon expression with lethal drug exposures by maintaining their epigenetic repression [121]. This evidence suggests combining HDACi with other targeted therapeutics may enhance their efficacy in treating cancer [128].

The examples mentioned above support the notion that retrotransposon activation in tumors may contribute to their turning into ‘hot tumors’, which are inflamed and T-cell- infiltrated tumors [169]. In such a microenvironment, the antitumor immune response will reduce the tumor burden and sensitize it to other targeted therapies and immunotherapy [169]. Retrotransposon activity in cancer probably occurs more in specific tumor types than in others [60,61]. It is unclear whether this is related to a more vigorous immune defense or a higher level of cellular adaptation by implementing changes in their epigenome or transcriptome [10].

Tumor-derived extracellular vesicles (EVs) are enriched in retrotransposon RNA and involved in the horizontal transfer of retrotransposons to normal cells. They may broadly influence the tumor microenvironment and immune response [170,171]. This evidence suggests that EVs facilitate the release and transfer of retrotransposons to other cells, contributing to tumor evolution or metastasis (if derived from tumor cells). Also, retrotransposon RNA transfer can influence recipient cells’ transcriptional and post-transcriptional regulation. For example, the increased L1-derived RNA transcripts in recipient cells after the EVs transfer activate members of the APOBEC3 [171]. EVs are currently subject to multiple clinical trials at different phases and are to be used as non-invasive tools for diagnosis and therapeutics. They can serve as cargo for drug delivery in cancer and other conditions (as referred to https://clinicaltrials.gov/, accessed on 18 June 2023). The increased expression of retrotransposons in EVs derived from tumor cells compared to those derived from normal cells [170] could potentially serve as a valuable biomarker for diagnostic purposes. Studies to characterize the origin, biogenesis, and destination of EVs containing retrotransposon RNA and protein in cancer patients are currently needed to understand their potential fully.

## 6. Closing Remarks

Overall, the advances in sequencing technologies and bioinformatic analysis made studying the activity of retrotransposons in cancer more accessible than before. However, these advances are accompanied by the complexities of dealing with big data. Therefore, tools are being developed to study retrotransposons to cope with these concerns and bring rigorous methods and strategies to keep the field moving forward.

Different cellular and molecular mechanisms regulate the activity of retrotransposons in the human genome. The deregulation of these mechanisms can activate retrotransposons and contribute to the process of tumorigenesis. Accumulating evidence indicates strong associations between retrotransposons and type I IFN immune responses. Retrotransposons could be carried in the extracellular space by tumor-derived EVs, which facilitate their release in the cytosol of surrounding cells, where different PRRs detect them. This detection can activate IRF-mediated type I IFN responses. An inflammatory response could be generated from IFN signaling, leading to a negative feedback loop to inhibit further retrotransposon activity (Figure 4). Extensive research to validate these assumptions is required in different types of tumors; this research is currently more accessible due to the advances in sequencing technologies and the strategies of bioinformatic data analysis.

Prospectively, these retrotransposon-induced inflammatory responses could be used as tools to improve options for cancer treatment by considering the variations between different types of cancer and tailoring the therapeutic choices to the associated response.

## Figures and Tables

**Figure 1 cancers-15-04340-f001:**
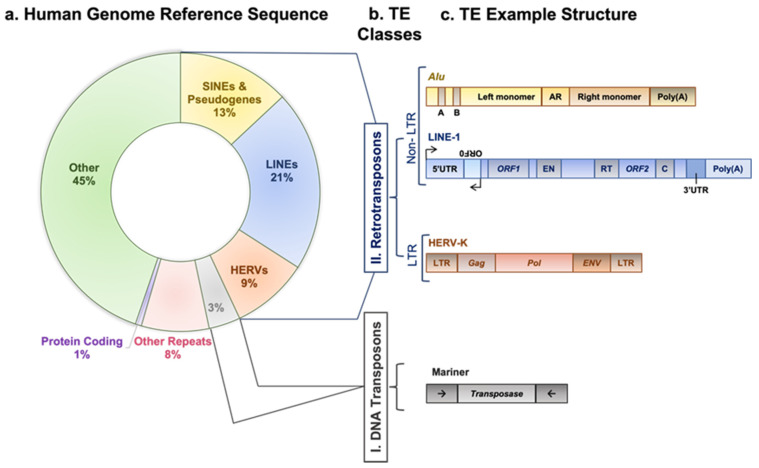
Transposable element classes, structure, and activity. (**a**) A doughnut chart represents fractions of human genome reference sequence constituents, as described in the recent telomere-to-telomere (T2T) assembly [6]. (**b**) TE main categories: first class, DNA transposons; second class, retrotransposons. The latter is subcategorized into elements having or lacking LTRs. LTR-containing elements include HERVs family, and non-LTR elements include SINEs, LINEs, and pseudogenes families. (**c**) Outline diagrams to represent structure of example elements per classes of transposable elements; *Alu* element from SINEs family is composed of two monomers separated by adenosine-rich (AR) linker. The left monomer contains an internal RNA polymerase III promoter (bars labeled A and B), and the right monomer is followed by a poly (A) tail. L1 element is a protein-coding element of the LINEs family; it has an internal promoter in its 5′ untranslated region (5′UTR) followed by a primate-specific antisense region (ORF0) and regions encoding L1 proteins (ORF1 and ORF2). ORF1p is a nuclear binding protein, and ORF2p has EN, RT, and cysteine-rich (C) domains. L1 element is ended by a poly (A) tail in its 3′ untranslated region (3′UTR). HERV-K element of the HERVs family contains two LTR regions separated by gag, pol, and env regions. Mariner of the DNA transposons class encodes transposase, an enzyme that binds and cuts near inverted repeats flanking the element (denoted by little arrows).

**Figure 2 cancers-15-04340-f002:**
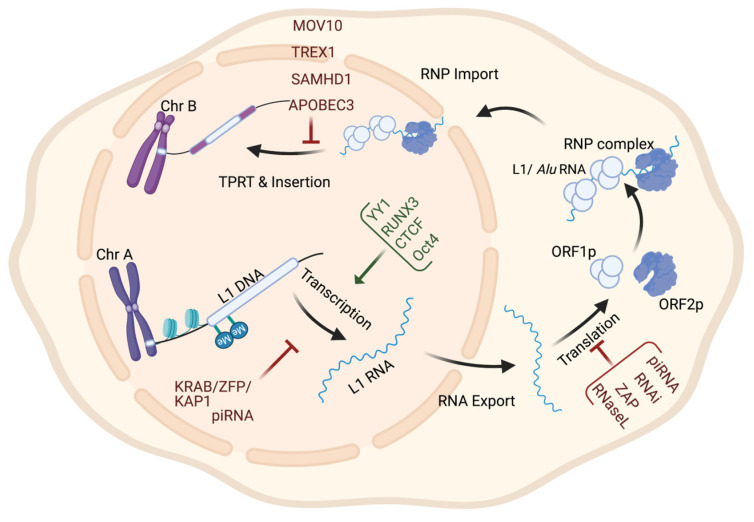
Retrotransposon levels of regulation throughout its life cycle. The regulation of retrotransposon activity can occur at the transcriptional level by histone modification or DNA methylation; at the post-transcriptional level by targeting RNA for degradation; and at the genomic insertion level by interfering with RNP complexes integrity or inhibiting TPRT.

**Figure 3 cancers-15-04340-f003:**
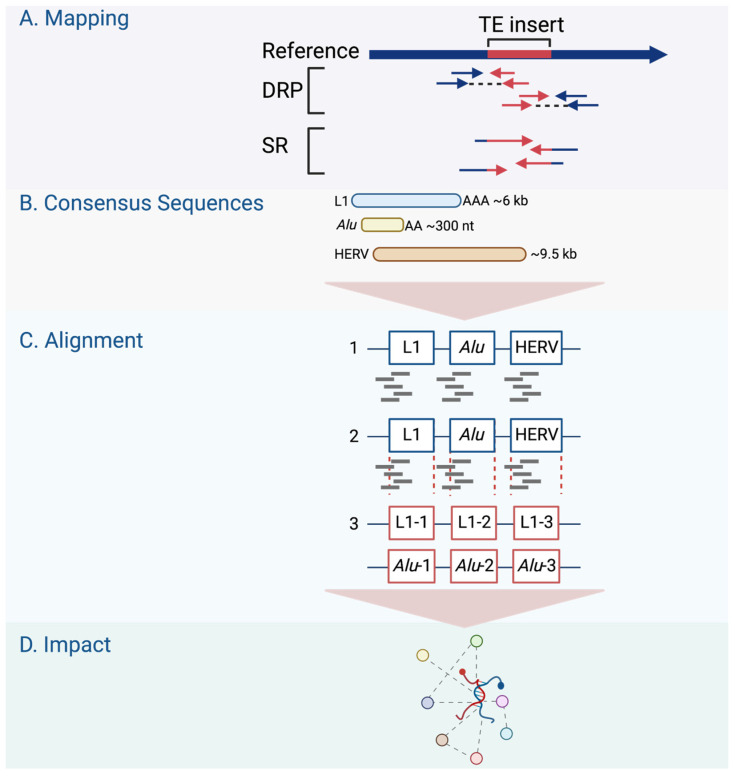
Genome-wide retrotransposon studies workflow. (**A**) TE novel insertions are detected in the human genome using sequencing reads using different mapping strategies, including DRP and SR. (**B**–**D**) The consensus sequences of active TE classes are obtained from repository databases and aligned to identify their differential expression in the genome.

**Figure 4 cancers-15-04340-f004:**
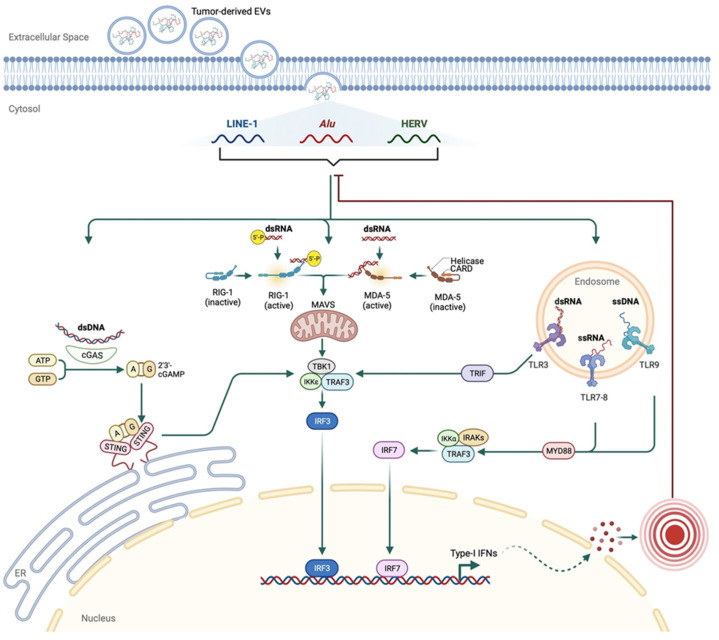
A proposed mechanism for retrotransposon-induced immune response in cancer. Tumor-derived EVs containing retrotransposons could be imported through the plasma membranes of other cells. Released cytosolic retrotransposons bind one of the PRRs, such as cGAS, RIG-I, MDA-5, or endosomal TLRs. The activated PRRs induce an IRF-mediated type I IFN response, stimulating inflammatory responses. These responses could create negative feedback on the retrotransposons to inhibit their activity.

**Table 1 cancers-15-04340-t001:** Mechanisms by which retrotransposons can affect the genome structure.

Retrotransposons Regulatory Effect	Citation	Schematic Illustration
Alternative promoter	[25,33,34]	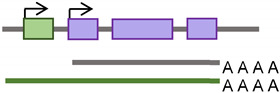
New enhancer/Silencer	[35,36]	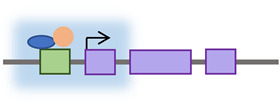
Exon disruption/addition	[37,38,39]	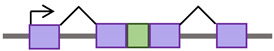
Alternative polyA	[40,41]	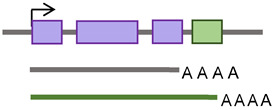
Regulatory RNA production	[42,43,44]	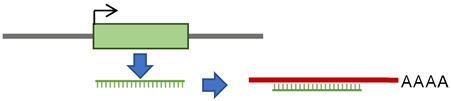
New Protein Production	[45]	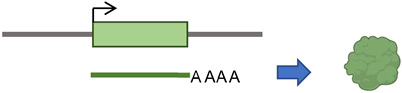
Alteration in splicing	[28,29,46]	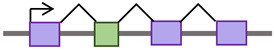
Deletion/duplication	[47,48]	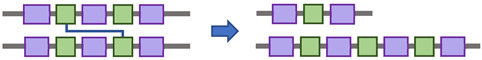
Insulation	[49,50]	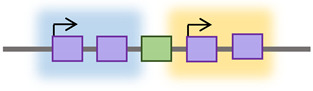

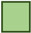
 Retrotransposon; 
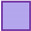
 Gene; 
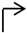
 Promoter; 
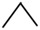
 Splicing; 
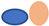
 Transcription Factors; **AAAA** PolyA tail; 
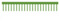
 RNA.

**Table 2 cancers-15-04340-t002:** Regulators of retrotransposon activity and their mechanism of regulation.

Regulator	Examples	Regulation Level	Regulation Mechanism	Study Model	Citations
Transcription factors	YY1, RUNX3, p53, Oct4, Sox2, Nanog, KLF4, MYC, CTCF, and BRCA1	Nuclear	Retrotransposon promoter binding and transcription activation.	HeLa NTeraD1 143B HCT116 HEPG2 hESCs MCF-7 K-562 GM12878 HEK-293 ES2	[75,76,77,78,79,80,81,82,83,86]
DNA methyltransferase enzymes	DNMT	Nuclear	DNA methylation of CpGs (retrotransposons contain ~half of CpG islands in the human genome).	Genome browser analysis	[87,88]
Histone marks	H3K9me3 and H3K27me3	Nuclear	Suppressive histone modifications associated with heterochromatin and frequently found on nucleosomes at TE loci.	147 cell types and ENCODE data	[89]
KRAB-ZFP/KAP1 complex		Nuclear	Transcriptional regulation of retrotransposons by inducing heterochromatin formation in somatic cells and promoting DNA methylation in early embryonic cells.	Human and mouse ESCs	[90,91]
Cytosine deaminases	AID, APOBEC1, APOBEC2, APOBEC3, and APOBEC4	Nuclear/ Cellular	Antiviral factors act to restrict retrotransposon by deaminating cytosine to uracil within DNA and RNA molecules or by physically interacting with retrotransposon RT to interfere with DNA polymerization during TPRT and target RNP complexes for sequestration in stress granules (SGs).	LLC-Mk2, Huh-7, HEK-293, HeLa, and U2OS cells	[92,93,94,95,96]
Aicardi–Goutières syndrome-associated genes	SAMHD1 and TREX1	Nuclear/ Cellular	Part of anti-retroviral response, SAMHD1 interacts directly with ORF2p in L1 RNP complexes. TERX1 interacts with ORF1p to change its subcellular localization and triggers its depletion.	HEK 293T, HeLa, and U2OS cells	[97,98]
Piwi-interacting RNA (piRNA)		Nuclear/ Cellular	These can form piRNA-induced silencing complex (piRISC), which allows PIWI proteins to specifically recognize and cleave retrotransposon transcripts by PIWI. PIWI proteins and piRNAs can also mediate CpG DNA methylation of retrotransposon promoters.	Mouse ESCs and D. melanogaster model	[99,100]
Antiviral response elements	MOV10, RNase L, and ZAP	Cellular	MOV10 sequesters L1 RNP and degrades L1 RNAs in SGs and cytoplasmic processing bodies (P-bodies). RNase L targets L1 RNA for degradation by an unknown mechanism. ZAP prevents the accumulation of L1 mRNA in the cytoplasm by targeting it to SGs.	HeLa, HEK 293T, and SW982 cells	[101,102,103]

**Table 3 cancers-15-04340-t003:** Retrotransposon activity in cancer genome-wide studies.

Citation	Data Used (Database)	Sample Size	Strategy	Focus	Important Findings
[61]	WGS (TCGA)	43	TE analyze (DRP reads)	Identifying novel insertions	One hundred and ninety-four somatic TE insertions in tumors, biased toward hypomethylated regions. Tumors of epithelial origin showed more pronounced L1 activity than brain and blood cancer types.
[64]	generated data	19	RC-seq	Identifying novel insertions	L1-mediated mechanisms enabling tumorigenesis in hepatocellular carcinoma, identified insertions in *MCC* and *ST18.*
[69]	WGS (TCGA and ICGC)	244	TraFiC pipeline (DRP reads)	Insertion characteristics and impact	A total of 2756 L1 somatic insertions in tumors, with colorectal and lung cancers being the most affected. Insertions exhibited hypomethylated promotors by tracking down their sources. L1 insertions demonstrated minimal to no effect on the course of tumorigenesis.
[66]	WGS, WES (TCGA)	967	TranspoSeq (DRP and SR reads)	Identifying novel insertions	Eight hundred and ten somatic retrotransposon insertions in epithelial cancers; many of them occurred in known cancer genes (by WGS). Thirty-five novel somatic retrotransposon insertions (by WES), including an insertion into an exon of the *PTEN*.
[64]	WGS	11	MELT (DRP and SR reads)	Identifying novel insertions	Hot L1 insertion in *APC* gene in colon cancer.
[116]	generated data	30	RC-seq	Identifying novel insertions	Eighty-eight tumor-specific L1 insertions in ovarian tumors; one intronic insertion added a novel cis-enhancer to *STC1* gene and promoted chemoresistance in cells bearing this mutation.
[117]	generated data	35 patients, 10 mice	RC-seq	Identifying novel insertions	First report of L1 activity in HCC murine tumors, identified 8 L1 tumor-specific insertions in 25 patients with alcohol abuse and 3 L1 insertions in 10 intra-hepatic cholangiocarcinoma patients.
[118]	WGS, RNA-seq (TCGA, EGA, dbGaP)	298	Modified TE analyzer (DRP reads)	Identifying novel insertions and impact	L1 activity positively associated with *TP53 mutation*. L1 insertion in exon of MOV10. Low L1 activity in tumors with high immune signature.
[119]	generated data	28	ATLAS-Seq	Characteristics of L1 integration	L1 shows a broad capacity for integration into all chromatin states compared to other mobile elements. L1 integration is influenced by the replication timing of target regions; distribution of new L1 insertions differs from those of pre-existing L1 elements.
[120]	WGS, RNA-seq (PCAWG)	2954	TraFiC pipeline (DRP reads)	Impact of insertions on structural variation	A total of 19,166 somatically acquired retrotransposition events that affected 35% of samples. L1 induced somatic structural variation in esophageal adenocarcinoma and was the second most frequent in head and neck and colorectal cancers.
[86]	WGS, RNA-seq (TCGA and GDC)	WGS: 54 ovarian cancers (OVCA) and matched normal. RNA-seq: 379 OVCA and 486 breast cancers.	MELT, RepEnrich, and Bayesian correlation	Identifying causes and consequences of retrotransposon expression in ovarian and breast cancer	Observed divergent inflammatory responses associated with retrotransposon expression in ovarian and breast cancer. Identified new factors inducing expression of endogenous retrotransposons such as anti-viral responses and the tumor suppressor BRCA1.

**Table 4 cancers-15-04340-t004:** Retrotransposon activity and associated immune response in cancer.

Citation	Model Used	TE Class	Type of Immune Response	Results Summary
[135]	hTERT1604, HCT116, SKMEL Cells	HERV and L1	Innate immune response to viral infection via dsRNA sensing pathway. Indirect T cell signaling	UHRF1 is required to suppress retrotransposon expression in human cells independently of DNA methylation. The downregulation of UHRF1 activated strong innate immune signaling, as confirmed by its restoration.
[136]	HEK293T, U87MG, THP-1, A549 cells	*Alu* and L1	Innate immune response to viral infection via MDA5	Constitutive activation of MDA5 (gain-of-function mutation) results from the loss of tolerance to cellular dsRNAs formed by *Alu*. *Alu:Alu* hybrids activate wild-type MDA5 under the ADAR1 deficiency.
[137]	Healthy donors’ PBMCs, PDACs	HERV and LINEs	Homeostatic and/or IFN-activated ISGs	Infection of tumor cells with H-1PV oncolytic virus is associated with a profound inhibition of TEs and innate immunity.
[138]	AML human cell lines	HERV and LINEs	Innate immune response via dsRNA-sensing pathway	Loss of SETDB1 gene in AML activates TEs which produce dsRNAs and trigger type I IFN response and apoptosis.
[139]	HEK293T	L1	Innate immune response	MDA5 directly binds to L1 5′-UTR and suppresses its promoter activity and inhibits its retrotransposition.
[118]	TCGA data of colorectal, stomach, and esophageal cancers	L1	Innate and adaptive immune response TLR and/or STAT6 signaling	GI tumors with high immune activity (e.g., those with EBV infection) carry a low number of L1 insertions and high levels of L1 suppressors (APOBEC3s and SAMHD1). Negative correlation between L1 regulatory T cells and PD1 signaling.
[140]	HEK 293T and 2102EP cells	L1	Innate immunity via TRIM5α	TRIM5α repress L1 activity by interacting with its RNPs in the cytoplasm. This interaction induces innate immune signaling via AP-1 and NF-κB to inhibit L1 promoter activity.
[141]	A549, MDCK, HEK 293T, and TZM-bl cells	HERV, LINE, and SINE	Innate immunity via TRIM28/KAP1	Influenza virus-triggered loss of SUMO-modified TRIM28, activates retrotransposons. Released cytosolic dsRNA induced IFN-mediated defense pathway.
[142]	Neuroblastoma transgenic mouse model, 4T1 cells	L1, SINE, and HERV	NF-κb and type I IFN inflammatory pathways	L1 de-silencing promoted drug resistance and activated IFN signaling. The use of NRTI reversed these phenotypes.
[143]	CMML and AML patients	LINE, SINE, HERV	Type I IFN pathway	DNMTi-treated samples presented TEs activation and IFN response triggering.
[144]	H69 cells and TCGA data	HERV	Innate immune signaling via MAVS and STING adaptive immune response	Mesenchymal tumor subpopulations trigger expression of a specific set of ERVs when exposed to IFNγ.
[145]	HT-29, HEK293T, and HeLa	HERV	Innate immune response via MDA5 and MAVS	ING3 loss decreased H3K27 trimethylation enrichment at HERVs.HERV activation induced IFN signaling.

## Data Availability

Not applicable.

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
