# Peer review of "The Regulation and Immune Signature of Retrotransposons in Cancer"

_cancers, 2023, doi:10.3390/cancers15174340_

Round 1

Reviewer 1 Report

In general, the manuscript is written well enough and contains a wealth of information in the form of references. Thus, the manuscript can be considered interesting for its field of study. The same aspect however exposes the weakness, in the sense that the manuscript seems to be limited to listing information but lacks a clear direction with strong conclusions.  Although correct in its current form, the manuscript is quite hard to get through due to its bulk, and does not really invite the reader to continue to the most relevant sections towards the end.

For this reason, the reviewer recommends re-editing the manuscript, a task which will not require anything but including data from literature (no experimentation, of course).

Comments:

-) A relatively small section of the manuscript actually deals with the immunological aspects of retrotransposons (principally, section 11 and to a lesser extend section 12, out of a total of 13 sections).  The bulk of the manuscript deals with regulation of retrotransposons and bioinformatics.  As a consequence, the content of the manuscript does not correspond to the title and abstract as provided. The use of “signature” in the title is very suggestive, too, but not covered by the main content.

This can be improved in two ways:

1)    Change the title and abstract. This would imply a change of scope and possibly the journal of submission.

2)    Extend the section on the immunological aspects. In particular, literature on innate immunity and IFN signaling, even connected to the accumulation of aberrant nucleic acids in the cytosplasm, is readily available. Concomitantly, the sections dealing with basic transposon biology can be shortened, in particular the section on bioinformatics analysis has little relation to immunity.

-) References are formatted in different ways; while some references have hyperlinks to the article, this is not present in others.  Please reformat.

-) Several paragraphs start with a statement of comparison “in contrast”.  The reader is left with the question in contrast to what (since a new paragraph is started).  The flow of the argument across paragraphs need to be checked and improved. Probably, moving paragraph breaks to a point where a more concluding sentence is used will work, although few strong conclusions are found in the manuscript.

All comments have been included in the previous section.

Reviewer 2 Report

Alkailani and Gibbings presents the review “The immune signature of retrotransposons in cancer”. They showed the advances in sequencing technologies and bioinformatic analysis of big data which facilitate studying the activity of jumping genes in the human genome from a broad perspective in cancer. I found the manuscript interesting, however, the content does not full meet the objective of the review. In addition, it is necessary to improve the writing because the grammatical structure is difficult to understand in several sections. I will list the weaknesses that, from my perspective, the manuscript presents and that could be considered for its modification and future submission. Several concerns must be fully addressed before potential publication of the manuscript.

1.    Homogenize the use of uppercase/lowercase letter of the subtopics.

2.    In subtopic #2 “Retrotransposon definition and classification” I suggest that the section be rewritten; perhaps paraphrase or improve the idea to make the information clearer. Consider an order; structure, function and mechanism.

3.    Place the name and number corresponding to table #1 exactly above it without the need to add a description since it is referred to in the text.

4.    In subtopic #4 “Mechanisms of retrotransposons regulation”.

-       I suggest describing some examples of each of the factors restricting retrotransposons at the nucleus and cytoplasm level (how is it that it restricts/interferes in the nucleus and how in the cytoplasm).

-       In Table 2, add a column where you mention what type of cancer is involved in the activity of the retrotransposons. Also, make the letter smaller or make adjustments to it so that the entire word fits in the table and is visually more understandable.

-       Figure 2.- Explain the levels of regulation of retrotransposons; LINE AND SIRE through their cycle of life (L1 and Alu, respectively).

5.    I suggest removing section 5 (Upon Defense Failure); I consider that this does not contribute anything to the manuscript.

6.    In subtopic 6: Retrotransposons in cancer.

-       Briefly explain the classical "two-hit" model.

-       I consider that the section needs to be strengthened by referring to the experimental evidence reported by other working groups.

-       I suggest that the authors should consider merging subtopic 3 with 6 with a possible subtopic proposal “Impact of retrotransposons on genome structure and function in cancer”.

7.    In subtopic 7 “Retrotransposons from a genome-wide perspective” I suggest that topic 2 and topic 7 be merged and can be titled “retrotranposon classification and a genome-wide perspective in cancer and mention examples of the retrotranposons of the immune response in cancer.

8.    In subtopic 8 “Bioinformatic tools to map retrotransposons”.

-       Mention prior to this subtopic, information that mentions the existing polymorphisms in transposable elements and their biological function in cancer.

-       Improve the size and quality of figure #3.

9.    Subtopic 8 and 9 (Bioinformatic tools to study the impact of retrotransposons) it is suggested that a single topic be made and the wording improved. Mention the bioinformatics tool you use, what each of these tools contributes, what is the utility and the strategies for the use of bioinformatics tools in the genome. Likewise, I suggest it be positioned after section 2 or as a subtopic of section 2.

10. In subtopic 10 “Immune signature of retrotransposons in cancer”.

-       Describe the evidence that mentions how retrotransposons regulate immune cells (for example: macrophages, lymphocytes, NKs, APCs, lymphocytes, among others) in the cancer immune response, what are the regulatory mechanisms they carry out.

-       Consider merging subtopic 10 with topic 11 (Another defense line against retrotransposon activity) and improve the description.

-       Mention examples of tumors that are not of viral origin compared to those of viral origin.

11. In subtopic 12 “Therapeutic opportunities for retrotransposon activity in cancer”.

-       First of all, mention the role of retrotranposons in cancer hallmarks and their importance in clinical application: I suggest restructuring this section.

-       In figure 4 they schematize the participation of extracellular vesicles, but it is not mentioned in any section of the text.

For the above, I consider that the content of this review is not sustainable for publication in its actual form. 

Moderate editing of English language required.

Round 2

Reviewer 2 Report

In the revised version of manuscript authors have replied all the concerns, thus I suggest to accept it for publication  in its actual form.